# Slalom: Fast, Verifiable and Private Execution of Neural Networks in Trusted Hardware

**Florian Tramèr**
Stanford University
tramer@cs.stanford.edu

**Dan Boneh**
Stanford University
dabo@cs.stanford.edu

## Abstract

As Machine Learning (ML) gets applied to security-critical or sensitive domains, there is a growing need for *integrity* and *privacy* for outsourced ML computations. A pragmatic solution comes from Trusted Execution Environments (TEEs), which use hardware and software protections to isolate sensitive computations from the untrusted software stack. However, these isolation guarantees come at a price in performance, compared to untrusted alternatives. This paper initiates the study of high performance execution of Deep Neural Networks (DNNs) in TEEs by efficiently partitioning DNN computations between trusted and untrusted devices. Building upon an efficient outsourcing scheme for matrix multiplication, we propose *Slalom*, a framework that securely delegates execution of *all linear layers* in a DNN from a TEE (e.g., Intel SGX or Sanctum) to a faster, yet untrusted, co-located processor. We evaluate Slalom by running DNNs in an Intel SGX enclave, which selectively delegates work to an untrusted GPU. For canonical DNNs (VGG16, MobileNet and ResNet variants) we obtain $6\times$ to $20\times$ increases in throughput for verifiable inference, and $4\times$ to $11\times$ for verifiable and private inference.

## 1 Introduction

Machine learning is increasingly used in sensitive decision making and security-critical settings. At the same time, the growth in both cloud offerings and software stack complexity widens the attack surface for ML applications. This raises the question of *integrity* and *privacy* guarantees for ML computations in untrusted environments, in particular for ML tasks *outsourced* by a client to a remote server. Prominent examples include cloud-based ML APIs (e.g., a speech-to-text application that consumes user-provided data) or general ML-as-a-Service platforms.

Trusted Execution Environments (TEEs), e.g, Intel SGX (McKeen et al., 2013), ARM TrustZone (Alves & Felton, 2004) or Sanctum (Costan et al., 2016) offer a pragmatic solution to this problem. TEEs use hardware and software protections to isolate sensitive code from other applications, while *attesting* to its correct execution. Running outsourced ML computations in TEEs provides remote clients with strong privacy and integrity guarantees.

For outsourced ML computations, TEEs outperform pure cryptographic approaches (e.g, (Gilad-Bachrach et al., 2016; Mohassel & Zhang, 2017; Ghodsi et al., 2017; Juvekar et al., 2018)) by multiple orders of magnitude. At the same time, the isolation guarantees of TEEs still come at a steep price in performance, compared to untrusted alternatives (i.e., running ML models on contemporary hardware with no security guarantees). For instance, Intel SGX (Intel Corp., 2015) incurs significant overhead for memory intensive tasks (Orenbach et al., 2017; Harnik & Tsfadia, 2017), has difficulties exploiting multi-threading, and is currently limited to desktop CPUs that are outmatched by untrusted alternatives (e.g., GPUs or server CPUs). Thus, our thesis is that *for modern ML workloads, TEEs will be at least an order of magnitude less efficient than the best available untrusted hardware.*

This leads us to the main question of this paper: *How can we most efficiently leverage TEEs for secure machine learning?* This was posed by Stoica et al. (2017) as one of nine open research problems for system challenges in AI. A specific challenge they raised is that of appropriately splitting ML computations between trusted and untrusted components, to increase efficiency as well as security by minimizing the Trusted Computing Base.

This paper explores a novel approach to this challenge, wherein a Deep Neural Network (DNN) execution is *partially outsourced from a TEE to a co-located, untrusted but faster device*. Our approach, inspired by the verifiable ASICs of Wahby et al. (2016), differs from cryptographic ML outsourcing. In our case, work is delegated between two *co-located* parties, thus allowing for highly interactive—yet conceptually simpler—outsourcing protocols with orders-of-magnitude better efficiency. Our work also departs from prior systems that execute DNNs fully in a TEE (Ohrimenko et al., 2016; Hunt et al., 2018; Cheng et al., 2018; Hanzlik et al., 2018).

The main observation that guides our approach is that matrix multiplication—the main bottleneck in DNNs—admits a concretely efficient verifiable outsourcing scheme known as Freivalds' algorithm (Freivalds, 1977), which can also be turned private in our setting. Our TEE selectively outsources these CPU intensive steps to a fast untrusted co-processor (and runs the remaining steps itself) therefore achieving much better performance than running the entire computation in the enclave, without compromising security.

**Contributions.** We propose Slalom, a framework for efficient DNN inference in *any* trusted execution environment (e.g., SGX or Sanctum). To evaluate Slalom, we build a lightweight DNN library for Intel SGX, which may be of independent interest. Our library allows for outsourcing all linear layers to an untrusted GPU without compromising integrity or privacy. Our code is available at `https://github.com/ftramer/slalom`.

We formally prove Slalom's security, and evaluate it on multiple canonical DNNs with a variety of computational costs—VGG16 (Simonyan & Zisserman, 2014), MobileNet (Howard et al., 2017), and ResNets (He et al., 2016). Compared to running all computations in SGX, outsourcing linear layers to an untrusted GPU increases throughput (as well as energy efficiency) by $6\times$ to $20\times$ for verifiable inference, and by $4\times$ to $11\times$ for verifiable and private inference. Finally, we discuss open challenges towards efficient verifiable training of DNNs in TEEs.

## 2 BACKGROUND

### 2.1 PROBLEM SETTING

We consider an outsourcing scheme between a client $\mathcal{C}$ and a server $\mathcal{S}$, where $\mathcal{S}$ executes a DNN $F(x) : \mathcal{X} \to \mathcal{Y}$ on data provided by $\mathcal{C}$. The DNN can either belong to the user (e.g., as in some ML-as-a-service platforms), or to the server (e.g., as in a cloud-based ML API). Depending on the application, this scheme should satisfy one or more of the following security properties (see Appendix B for formal definitions):

- **t-Integrity:** For any $\mathcal{S}$ and input $x$, the probability that a user interacting with $\mathcal{S}$ does not abort (i.e., output $\perp$) and outputs an incorrect value $\tilde{y} \neq F(x)$ is less than $t$.
- **Privacy:** The server $\mathcal{S}$ learns no information about the user's input $x$.
- **Model privacy:** If the model $F$ is provided by the user, $\mathcal{S}$ learns no information about $F$ (beyond e.g., its approximate size). If $F$ belongs to the server, $\mathcal{C}$ learns no more about $F$ than what is revealed by $y = F(x)$.[1]

### 2.2 TRUSTED EXECUTION ENVIRONMENTS (TEES), INTEL SGX, AND A STRONG BASELINE

Trusted Execution Environments (TEE) such as Intel SGX, ARM TrustZone or Sanctum (Costan et al., 2016) enable execution of programs in *secure enclaves*. Hardware protections isolate computations in enclaves from all

---

[1] For this zero-knowledge guarantee to be meaningful in our context, $\mathcal{S}$ would first *commit* to a specific DNN, and then convince $\mathcal{C}$ that this DNN was correctly evaluated on her input, without revealing anything else about the DNN.

Table 1: **Security guarantees and performance (relative to baseline) of different ML outsourcing schemes**.

| Approach | TEE | Integrity | Privacy | Model Privacy | | Throughput (relative) |
|---|---|---|---|---|---|---|
| | | | | w.r.t. Server | w.r.t. Client | |
| SafetyNets (Ghodsi et al., 2017) | - | ● | ○ | ○ | ○ | $\leq {}^1\!/_{200} \times$ |
| Gazelle (Juvekar et al., 2018) | - | ○ | ●[*] | ○ | ◖ | $\leq {}^1\!/_{1000} \times$ |
| **Secure baseline** (run DNN in TEE) | ✓ | ● | ● | ● | ● | $1\times$ |
| Insecure baseline (run DNN on GPU) | - | ○ | ○ | ○ | ◖ | $\geq 50\times$ |
| **Slalom (Ours)** | ✓ | ● | ●[*] | ○ | ● | $4\times$ - $20\times$ |

[*] With an offline preprocessing phase.

programs on the same host, including the operating system. Enclaves can produce *remote attestations*—digital signatures over an enclave's code—that a remote party can verify using the manufacturer's public key. Our experiments with Slalom use hardware enclaves provided by Intel SGX (see Appendix A for details).[2]

TEEs offer an efficient solution for ML outsourcing: The server runs an enclave that initiates a secure communication with $\mathcal{C}$ and evaluates a model $F$ on $\mathcal{C}$'s input data. This simple scheme (which we implemented in SGX, see Section 4) outperforms cryptographic ML outsourcing protocols by 2-3 orders of magnitude (albeit under a different trust model). See Table 1 and Appendix C for a comparison to two representative works.

Yet, SGX's security comes at a performance cost, and there remains a large gap between TEEs and untrusted devices. For example, current SGX CPUs are limited to 128 MB of *Processor Reserved Memory* (PRM) (Costan & Devadas, 2016) and incur severe paging overheads when exceeding this allowance (Orenbach et al., 2017). We also failed to achieve noticeable speed ups for multi-threaded DNN evaluations in SGX enclaves (see Appendix H). For DNN computations, current SGX enclaves thus cannot compete—in terms of performance or energy efficiency (see Appendix C)—with contemporary *untrusted* hardware, such as a GPU or server CPU.

In this work, we treat the above simple (yet powerful) TEE scheme as a baseline, and identify settings where we can still improve upon it. We will show that our system, Slalom, substantially outperforms this baseline when the server has access to the model $F$ (e.g., $F$ belongs to $\mathcal{S}$ as in cloud ML APIs, or $F$ is public). Slalom performs best for verifiable inference (the setting considered in SafetyNets (Ghodsi et al., 2017)). If the TEE can run some offline data-independent preprocessing (e.g., as in Gazelle (Juvekar et al., 2018)), Slalom also outperforms the baseline for *private* (and verifiable) outsourced computations in a later online phase. Such a two-stage approach is viable if user data is sent at irregular intervals yet has to be processed with high throughput when available.

## 2.3 OUTSOURCING OUTSOURCED DNNs AND FREIVALDS' ALGORITHM

Our idea for speeding up DNN inference in TEEs is to *further outsource* work from the TEE to a *co-located faster untrusted processor*. Improving upon the above baseline thus requires that the *combined* cost of doing work on the untrusted device and verifying it in the TEE be cheaper than evaluating the full DNN in the TEE.

Wahby et al. (2016; 2017) aim at this goal for *arbitrary computations* outsourced between co-located ASICs. The generic non-interactive proofs they use for integrity are similar to those used in SafetyNets (Ghodsi et al., 2017), which incur overheads that are too large to warrant outsourcing in our setting (e.g., Wahby et al. (2016) find that the technology gap between trusted and untrusted devices needs to be of over two decades for their scheme to break even). Similarly for privacy, standard cryptographic outsourcing protocols (e.g., (Juvekar et al., 2018)) are unusable in our setting as simply running the computation in the TEE is much more efficient (see Table 1).

---

[2]SGX has recently come under several side-channel attacks (Chen et al., 2018; Van Bulck et al., 2018). Intel is making firmware and hardware updates to SGX with the goal of preventing these attacks. In time, it is likely that SGX can be made sufficiently secure to satisfy the requirements needed for Slalom. Even if not, other enclave architectures are available, such as Sanctum for RISC-V (Costan et al., 2016) or possibly a separate co-processor for security operations.

To overcome this barrier, we design outsourcing protocols tailored to DNNs, leveraging two insights:

1. In our setting, the TEE is *co-located* with the server's faster untrusted processors, thus widening the design space to *interactive* outsourcing protocols with high communication but better efficiency.

2. The TEE always has knowledge of the model and can *selectively* outsource part of the DNN evaluation and compute others—for which outsourcing is harder—itself.

DNNs are a class of functions that are particularly well suited for selective outsourcing. Indeed, non-linearities—which are hard to securely outsource (with integrity or privacy)—represent a small fraction of the computation in a DNN so we can evaluate these in the TEE (e.g., for VGG16 inference on a single CPU thread, about $1.5\%$ of the computation is spent on non-linearities). In contrast, linear operators—the main computational bottleneck in DNNs—admit for a conceptually simple yet concretely efficient secure delegation scheme, described below.

**Integrity.** We verify integrity of outsourced linear layers using variants of an algorithm by Freivalds (1977).

**Lemma 2.1** (Freivalds). *Let $A$, $B$ and $C$ be $n \times n$ matrices over a field $\mathbb{F}$ and let $s$ be a uniformly random vector in $\mathbb{S}^n$, for $\mathbb{S} \subseteq \mathbb{F}$. Then, $\Pr[Cs = A(Bs) \mid C \neq AB] = \Pr[(C - AB)s = \mathbf{0} \mid (C - AB) \neq \mathbf{0}] \leq 1/|\mathbb{S}|$.*

The randomized check requires $3n^2$ multiplications, a significant reduction (both in concrete terms and asymptotically) over evaluating the product directly. The algorithm has no false negatives and trivially extends to rectangular matrices. Independently repeating the check $k$ times yields soundness error $1/|\mathbb{S}|^k$.

**Privacy.** Input privacy for outsourced linear operators could be achieved with linearly homomorphic encryption, but the overhead (see the micro-benchmarks in (Juvekar et al., 2018)) is too high to compete with our baseline (i.e., computing the function directly in the TEE would be faster than outsourcing it over encrypted data).

We instead propose a very efficient two-stage approach based on symmetric cryptography, i.e., an additive stream cipher. Let $f : \mathbb{F}^m \to \mathbb{F}^n$ be a linear function over a field $\mathbb{F}$. In an offline phase, the TEE generates a stream of *one-time-use pseudorandom elements* $r \in \mathbb{F}^m$, and pre-computes $u = f(r)$. Then, in the online phase when the remote client sends an input $x$, the TEE computes $\mathsf{Enc}(x) = x + r$ over $\mathbb{F}^m$ (i.e., a secure encryption of $x$ with a stream cipher), and outsources the computation of $f(\mathsf{Enc}(x))$ to the faster processor. Given the result $f(\mathsf{Enc}(x)) = f(x + r) = f(x) + f(r) = f(x) + u$, the TEE recovers $f(x)$ using the pre-computed $u$.

**Communication.** Using Freivalds' algorithm and symmetric encryption for each linear layer in a DNN incurs high interaction and communication between the TEE and untrusted co-processor (e.g., over 50MB per inference for VGG16, see Table 3). This would be prohibitive if they were not co-located. There are protocols with lower communication than repeatedly using Freivalds' ((Fiore & Gennaro, 2012; Thaler, 2013; Ghodsi et al., 2017)). Yet, these incur a high overhead on the prover in practice and are thus not suitable in our setting.

## 3  SLALOM

We introduce Slalom, a three-step approach for outsourcing DNNs from a TEE to an untrusted but faster device: (1) Inputs and weights are quantized and embedded in a field $\mathbb{F}$; (2) Linear layers are outsourced and verified using Freivalds' algorithm; (3) Inputs of linear layers are encrypted with a pre-computed pseudorandom stream to guarantee privacy. Figure 1 shows two Slalom variants, one to achieve integrity, and one to also achieve privacy.

We focus on feed-forward networks with fully connected layers, convolutions, separable convolutions, pooling layers and activations. Slalom can be extended to other architectures (e.g., residual networks, see Section 4.3).

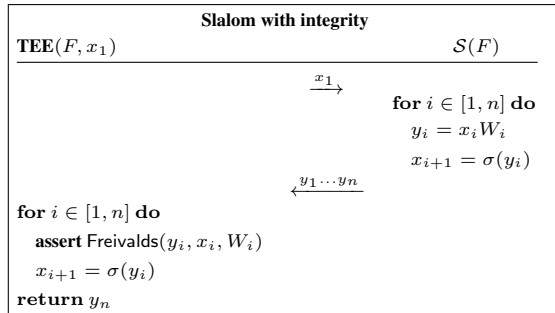

Figure 1: **The Slalom algorithms for verifiable and private DNN inference.** The TEE outsources computation of $n$ linear layers of a model $F$ to the untrusted host server $\mathcal{S}$. Each linear layer is defined by a matrix $W_i$ of size $m_i \times n_i$ and followed by an activation $\sigma$. All operations are over a field $\mathbb{F}$. The Freivalds$(y_i, x_i, w_i)$ subroutine performs $k$ repetitions of Freivalds' check (possibly using precomputed values as in Section 3.2). The pseudorandom elements $r_i$ (we omit the PRNG for simplicity) and precomputed values $u_i$ are used only once.

### 3.1 QUANTIZATION

The techniques we use for integrity and privacy (Freivalds' algorithm and stream ciphers) work over a field $\mathbb{F}$. We thus *quantize* all inputs and weights of a DNN to integers, and embed these integers in the field $\mathbb{Z}_p$ of integers modulo a prime $p$ (where $p$ is larger than all values computed in a DNN evaluation, so as to avoid wrap-around).

As in (Gupta et al., 2015), we convert floating point numbers $x$ to a *fixed-point representation* as $\tilde{x} = \mathrm{FP}(x; l) := \mathrm{round}(2^l \cdot x)$. For a linear layer with kernel $W$ and bias $b$, we define integer parameters $\tilde{W} = \mathrm{FP}(W, l)$, $\tilde{b} = \mathrm{FP}(b, 2l)$. After applying the layer to a quantized input $\tilde{x}$, we scale the output by $2^{-l}$ and re-round to an integer.

For efficiency reasons, we perform integer arithmetic using floats (so-called fake quantization), and choose $p < 2^{24}$ to avoid loss of precision (we use $p = 2^{24} - 3$). For the models we evaluate, setting $l = 8$ for all weights and inputs ensures that all DNN values are bounded by $2^{24}$, with less than a $0.5\%$ drop in accuracy (see Table 3). When performing arithmetic modulo $p$ (e.g., for Freivalds' algorithm or when computing on encrypted data), we use double-precision floats, to reduce the number of modular reductions required (details are in Appendix F).

### 3.2 VERIFYING COMMON LINEAR OPERATORS

We now describe Slalom's approach to verifying the integrity of outsourced linear layers. We describe these layers in detail in Appendix D and summarize this section's results in Table 2.

**Freivalds' Algorithm for Batches.** The most direct way of applying Freivalds' algorithm to arbitrary linear layers of a DNN is by exploiting batching. Any linear layer $f(x)$ from inputs of size $m$ to outputs of size $n$ can be represented (with appropriate reshaping) as $f(x) = x^\top W$ for a (often sparse and implicit) $m \times n$ matrix $W$.

For a batch $X$ of size $B$, we can outsource $f(X)$ and check that the output $Y$ satisfies $f(s^\top X) = s^\top Y$, for a random vector $s$ (we are implicitly applying Freivalds to the matrix product $XW = Y$). As the batch size $B$ grows, the cost of evaluating $f$ is amortized and the total verification cost is $|X| + |Y| + \mathrm{cost}_f$ multiplications (i.e., we approach one operation per input and output). Yet, as we show in Section 4.3, while batched verification is worthwhile for processors with larger memory, it is prohibitive in SGX enclaves due to the limited PRM.

For full convolutions (and pointwise convolutions), a direct application of Freivalds' check is worthwhile even for single-element batches. For $f(x) = \mathrm{Conv}(x, W)$ and purported output $y$, we can sample a random vector $s$ of

Table 2: **Complexity (number of multiplications) for evaluating and verifying linear functions.** The layers are "Fully Connected", "Convolution", "Depthwise Convolution" and "Pointwise Convolution", defined in Appendix D. Each layer $f$ has an input $x$, output $y$ and kernel $W$. We assume a batch size of $B \geq 1$.

| Layer | $|\mathbf{x}|, |\mathbf{y}|$ | $|\mathbf{W}|$ | $\mathsf{cost}_f$ ($\mathbf{B} = 1$) | Batched verification | With preproc. |
|---|---|---|---|---|---|
| **FC** | $h_{\text{in}}, h_{\text{out}}$ | $h_{\text{in}} \cdot h_{\text{out}}$ | $|x| \cdot |y|$ | $B \cdot (|x| + |y|) + \mathsf{cost}_f$ | $B \cdot (|x| + |y|)$ |
| **Conv** | $h \cdot w \cdot c_{\text{in}}, h \cdot w \cdot c_{\text{out}}$ | $k^2 \cdot c_{\text{in}} \cdot c_{\text{out}}$ | $|x| \cdot k^2 \cdot c_{\text{out}}$ | $B \cdot (|x| + |y|) + c_{\text{in}} \cdot c_{\text{out}} + |x| \cdot k^2$ | $B \cdot (|x| + |y|)$ |
| **Depth. Conv** | $h \cdot w \cdot c_{\text{in}}, h \cdot w \cdot c_{\text{in}}$ | $k^2 \cdot c_{\text{in}}$ | $|x| \cdot k^2$ | $B \cdot (|x| + |y|) + \mathsf{cost}_f$ | $B \cdot (|x| + |y|)$ |
| **Point. Conv** | $h \cdot w \cdot c_{\text{in}}, h \cdot w \cdot c_{\text{out}}$ | $c_{\text{in}} \cdot c_{\text{out}}$ | $|x| \cdot c_{\text{out}}$ | $B \cdot (|x| + |y|) + c_{\text{in}} \cdot c_{\text{out}}$ | $B \cdot (|x| + |y|)$ |

dimension $c_{\text{out}}$ (the number of output channels), and check that $\text{Conv}(x, Ws) = ys$ (with appropriate reshaping). For a batch of inputs $X$, we can also apply Freivalds' algorithm twice to reduce both $W$ and $X$.

**Preprocessing.** We now show how to obtain an outsourcing scheme for linear layers that has optimal verification complexity (i.e., $|x| + |y|$ operations) for single-element batches and arbitrary linear operators, while at the same time *compressing* the DNN's weights (a welcome property in our memory-limited TEE model).

We leverage two facts: (1) DNN weights are fixed at inference time, so part of Freivalds' check can be pre-computed; (2) the TEE can keep secrets from the host $\mathcal{S}$, so the random values $s$ can be re-used across layers or inputs (if we run Freivalds' check $n$ times with the same secret randomness, the soundness errors grows at most by a factor $n$). Our verification scheme with preprocessing follows from a reformulation of Lemma (2.1):

**Lemma 3.1.** *Let* $f : \mathbb{F}^m \to \mathbb{F}^n$ *be a linear operator,* $f(x) \coloneqq x^\top W$. *Let* $s$ *be uniformly random in* $\mathbb{S}^n$, *for* $\mathbb{S} \subseteq \mathbb{F}$, *and let* $\tilde{s} \coloneqq \nabla F_x(s) = Ws$. *For any* $x \in \mathbb{F}^m$, $y \in \mathbb{F}^n$, *we have* $\Pr\left[y^\top s = x^\top \tilde{s} \mid y \neq f(x)\right] \leq 1/|\mathbb{S}|$.

The check requires $|x| + |y|$ multiplications, and storage for $s$ and $\tilde{s} \coloneqq Ws$ (of size $|x|$ and $|y|$). To save space, we can reuse the same random $s$ for every layer. The memory footprint of a model is then equal to the size of the inputs of all its linear layers (e.g., for VGG16 the footprint is reduced from 550MB to 36MB, see Table 3).

### 3.3 INPUT PRIVACY

To guarantee privacy of the client's inputs, we use precomputed blinding factors for each outsourced computation, as described in Section 2.3. The TEE uses a cryptographic Pseudo Random Number Generator (PRNG) to generate blinding factors. The precomputed "unblinding factors" are encrypted and stored in untrusted memory or disk. In the online phase, the TEE regenerates the blinding factors using the same PRNG seed, and uses the precomputed unblinding factors to decrypt the output of the outsourced linear layer.

This blinding process incurs several overheads: (1) the computations on the untrusted device have to be performed over $\mathbb{Z}_p$ so we use double-precision arithmetic. (2) The trusted and untrusted processors exchange data in-between each layer, rather than at the end of a full inference pass. (3) The TEE has to efficiently load precomputed unblinding factors, which requires either a large amount of RAM, or a fast access to disk (e.g., a PCIe SSD).

Slalom's security is given by the following results. Formal definitions and proofs are in Appendix B. Let negl be a *negligible* function (for any integer $c > 0$ there exists an integer $N_c$ such that for all $x > N_c$, $|\text{negl}(x)| < 1/x^c$).

**Theorem 3.2.** *Let* Slalom *be the protocol from Figure 1 (right), where* $F$ *is an* $n$-*layer DNN, and Freivalds' algorithm is repeated* $k$ *times per layer with random vectors drawn from* $\mathbb{S} \subseteq \mathbb{F}$. *Assume all random values are generated using a secure PRNG with security parameter* $\lambda$. *Then,* Slalom *is a secure outsourcing scheme for* $F$ *between a TEE and an untrusted co-processor* $\mathcal{S}$ *with privacy and* $t$-*integrity for* $t = n/|\mathbb{S}|^k - \text{negl}(\lambda)$.

**Corollary 3.3.** *Assuming the TEE is secure (i.e., it acts as a trusted third party hosted by* $\mathcal{S}$), Slalom *is a secure outsourcing scheme between a remote client* $\mathcal{C}$ *and server* $\mathcal{S}$ *with privacy and* $t$-*integrity for* $t = n/|\mathbb{S}|^k - \text{negl}(\lambda)$. *If the model* $F$ *is the property of* $\mathcal{S}$, *the scheme further satisfies model privacy.*

## 4 EMPIRICAL EVALUATION

We evaluate Slalom on real Intel SGX hardware, on micro-benchmarks and a sample application (ImageNet inference with VGG16, MobileNet and ResNet models). Our aim is to show that, compared to a baseline that runs inference fully in the TEE, outsourcing linear layers increases performance without sacrificing security.

### 4.1 IMPLEMENTATION

As enclaves cannot access most OS features (e.g., multi-threading, disk and driver IO), porting a large framework such as TensorFlow or Intel's MKL-DNN to SGX is hard. Instead, we designed a lightweight C++ library for feed-forward networks based on Eigen, a linear-algebra library which TensorFlow uses as a CPU backend. Our library implements the forward pass of DNNs, with support for dense layers, standard and separable convolutions, pooling, and activations. When run on a native CPU (without SGX), its performance is comparable to TensorFlow on CPU (compiled with AVX). Our code is available at `https://github.com/ftramer/slalom`.

Slalom performs arithmetic over $\mathbb{Z}_p$, for $p = 2^{24} - 3$. For integrity, we apply Freivalds' check twice to each layer ($k = 2$), with random values from $\mathbb{S} = [-2^{19}, 2^{19}]$, to achieve 40 bits of *statistical soundness* per layer (see Appendix F for details on the selection of these parameters). For a 50-layer DNN, $\mathcal{S}$ has a chance of less than 1 in 22 billion of fooling the TEE on any incorrect DNN evaluation (a slightly better guarantee than in SafetyNets). For privacy, we use AES-CTR and AES-GCM to generate, encrypt and authenticate blinding factors.

### 4.2 SETUP

We use an Intel Core i7-6700 Skylake 3.40GHz processor with 8GB of RAM, a desktop processor with SGX support. The outsourced computations are performed on a co-located Nvidia TITAN XP GPU. Due to a lack of native internal multi-threading in SGX, we run our TEE in a single CPU thread. We discuss challenges for efficient parallelization in Appendix H. We evaluate Slalom on the following workloads:

- Synthetic benchmarks for matrix products, convolutions and separable convolutions, where we compare the enclave's running time for computing a linear operation to that of solely verifying the result.
- ImageNet (Deng et al., 2009) classification with VGG16 (Simonyan & Zisserman, 2014), MobileNet (Howard et al., 2017), and ResNet He et al. (2016) models (with fused Batch Normalization layers when applicable).

MobileNet, a model tailored for low compute devices, serves as a worst-case benchmark for Slalom, as the model's design aggressively minimizes the amount of computation performed per layer. We also consider a "fused" variant of MobileNet with no activation between depthwise and pointwise convolutions. Removing these activations *improves* convergence and accuracy (Chollet, 2017; Sheng et al., 2018), while also making the network more outsourcing-friendly (i.e., it is possible to verify a separable convolution in a single step).

Our evaluation focuses on throughput (number of forward passes per second). We also discuss energy efficiency in Appendix C to account for hardware differences between our baseline (TEE only) and Slalom (TEE + GPU).

### 4.3 RESULTS

**Micro-Benchmarks.** Our micro-benchmark suite consists of square matrix products of increasing dimensions, convolutional operations performed by VGG16, and separable convolutions performed by MobileNet. In all cases, the data is pre-loaded inside an enclave, so we only measure the in-enclave execution time. Figure 2 plots the relative speedups of various verification strategies over the cost of computing the linear operation directly. In all cases, the baseline computation is performed in single-precision floating point, and the verification algorithms repeat Freivalds' check so as to attain at least 40 bits of statistical soundness.

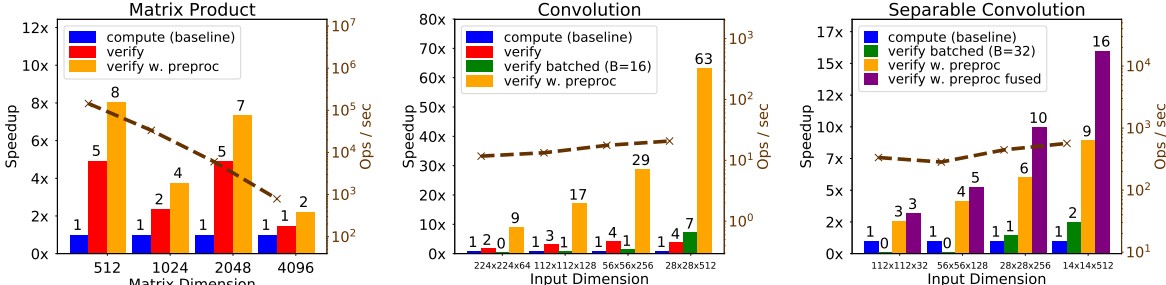

Figure 2: **Micro benchmarks on Intel SGX.** We plot the relative speedup of verifying the result of a linear operator compared to computing it entirely in the enclave. The dotted line shows the throughput obtained for a direct computation. "Fused" separable convolutions contain no intermediate activation.

For square matrices of dimensions up to 2048, verifying an outsourced result is $4\times$ to $8\times$ faster than computing it. For larger matrices, we exceed the limit of SGX's DRAM, so the enclave resorts to expensive paging which drastically reduces performance both for computation and verification.

For convolutions (standard or separable), we achieve large savings with outsourcing if Freivalds' algorithm is applied with preprocessing. The savings get higher as the number of channels increases. Without preprocessing, Freivalds' algorithm results in savings when $c_{out}$ is large. Due to SGX's small PRM, batched verification is only effective for operators with small memory footprints. As expected, "truly" separable convolutions (with no intermediate non-linearity) are much faster to verify, as they can be viewed as a single linear operator.

**Verifiable Inference.** Figure 3 shows the throughout of end-to-end forward passes in two neural networks, VGG16 and MobileNet. For integrity, we compare the secure baseline (executing the DNN fully in the enclave) to two variants of the Slalom algorithm in Figure 1. The first (in red) applies Freivalds' algorithm "on-the-fly", while the second more efficient variant (in orange) pre-computes part of Freivalds' check as described in Section 3.2.

The VGG16 network is much larger (500MB) than SGX's PRM. As a result, there is a large overhead on the forward pass and verification without preprocessing. If the enclave securely stores preprocessed products $Wr$ for all network weights, we drastically reduce the memory footprint and achieve up to a $20.3\times$ increase in throughput. We also ran the lower-half of the VGG16 network (without the fully connected layers), a common approach for extracting features for transfer learning or object recognition (Liu et al., 2016). This part fits in the PRM, and we thus achieve higher throughput for in-enclave forward passes and on-the-fly verification.

For MobileNet, we achieve between $3.6\times$ and $6.4\times$ speedups when using Slalom for verifiable inference (for the standard or "fused" model, respectively). The speedups are smaller than for VGG16, as MobileNet performs much fewer operations per layer (verifying a linear layer requires computing at least two multiplications for each input and output. The closer the forward pass gets to that lower-bound, the less we can save by outsourcing).

**Private Inference.** We further benchmark the cost of private DNN inference, where inputs of outsourced linear layers are additionally *blinded*. Blinding and unblinding each layer's inputs and outputs is costly, especially in SGX due to the extra in-enclave memory reads and writes. Nevertheless, for VGG16 and the fused MobileNet variant without intermediate activations, we achieve respective speedups of $13.0\times$ and $5.0\times$ for private outsourcing (in black in Figure 3), and speedups of $10.7\times$ and $4.1\times$ when also ensuring integrity (in purple). For this benchmark, the precomputed unblinding factor are stored in untrusted memory.

We performed the same experiments on a standard CPU (i.e., without SGX) and find that Slalom's improvements are even higher in non-resource-constrained or multi-threaded environments (see Appendix G-H). Slalom's improvements over the baseline also hold when accounting for energy efficiency (see Section C).

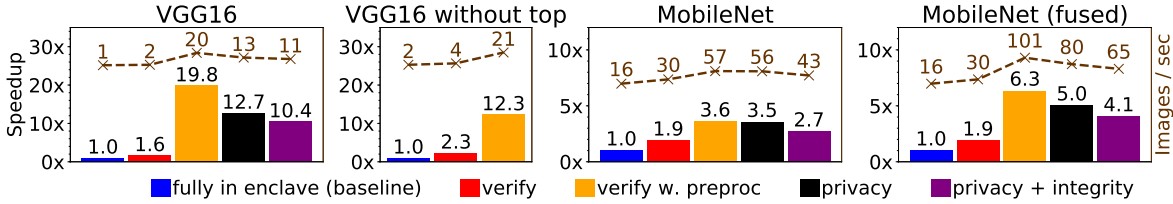

Figure 3: **Verifiable and private inference with Intel SGX.** We show results for VGG16, VGG16 without the fully connected layers, MobileNet, and a fused MobileNet variant with no intermediate activation for separable convolutions. We compare the baseline of fully executing the DNN in the enclave (blue) to different secure outsourcing schemes: integrity with Freivalds (red); integrity with Freivalds and precomputed secrets (yellow); privacy only (black); privacy and integrity (purple).

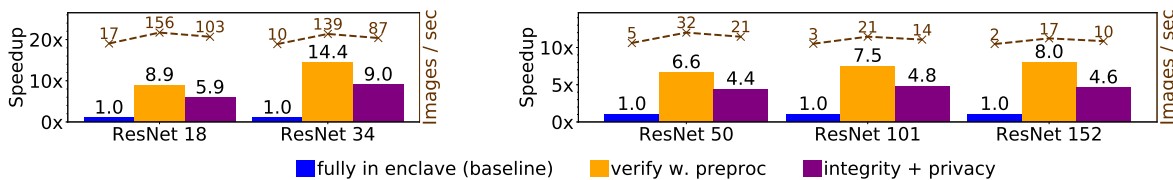

Figure 4: **Secure outsourcing of ResNet models with Intel SGX.** We compare the baseline of fully executing the DNN in the enclave (blue) to secure outsourcing with integrity (yellow) and privacy and integrity (purple).

**Extending Slalom to Deep Residual Networks.** The Slalom algorithm in Figure 1 and our evaluations above focus on feed-forward architectures. Extending Slalom to more complex DNNs is quite simple. To illustrate, we consider the family of ResNet models (He et al., 2016), which use residual blocks $f(x) = \sigma(f_1(x) + f_2(x))$ that merge two feed-forward "paths" $f_1$ and $f_2$ into a final activation $\sigma$. To verify integrity of $f(x)$, the TEE simply verifies all linear layers in $f_1$ and $f_2$ and computes $\sigma$ directly. For privacy, the TEE applies the interactive Slalom protocol in Figure 1 (right) in turn to $f_1$ and $f_2$, and then computes $\sigma$. The results for the *privacy-preserving* Slalom variant in Figure 4 use a preliminary implementation that performs all required operations—and thus provides meaningful performance numbers—but without properly constructed unblinding factors.

We use the ResNet implementation from Keras Chollet et al. (2015), which contains a pre-trained 50-layer variant. For this model, we find that our quantization scheme results in less than a $0.5\%$ decrease in accuracy (see Table 3). For other variants (i.e., with $18, 34, 101$ and $152$ layers) we compute throughput on untrained models. Figure 4 shows benchmarks for different ResNet variants when executed fully in the enclave (our baseline) as well as secure outsourcing with integrity or privacy and integrity. For all models, we achieve $6.6\times$ to $14.4\times$ speedups for verifiable inference and $4.4\times$ to $9.0\times$ speedups when adding privacy.

Comparing results for different models is illustrative of how Slalom's savings scale with model size and architectural design choices. The $18$ and $34$-layer ResNets use convolutions with $3 \times 3$ kernels, whereas the larger models mainly use pointwise convolutions. As shown in Table 2 verifying a convolution is about a factor $k^2 \cdot c_{out}$ than computing it, which explains the higher savings for models that use convolutions with large kernel windows. When adding more layers to a model, we expect Slalom's speedup over the baseline to remain constant (e.g., if we duplicate each layer, the baseline computation and the verification should both take twice as long). Yet we find that Slalom's speedups usually *increase* as layers get added to the ResNet architecture. This is because the deeper ResNet variants are obtained by duplicating layers towards the end of the pipeline, which have the largest number of channels and for which Slalom achieves the highest savings.

## 5 Challenges for Verifiable and Private Training

Our techniques for secure outsourcing of DNN inference might also apply to DNN training. Indeed, a backward pass consists of similar linear operators as a forward pass, and can thus be verified with Freivalds' algorithm. Yet, applying Slalom to DNN training is challenging, as described below, and we leave this problem open.

- Quantizing DNNs for training is harder than for inference, due to large changes in weight magnitudes (Micikevicius et al., 2018). Thus, a more flexible quantization scheme than the one we used would be necessary.
- Because the DNN's weights change during training, the same preprocessed random vectors for Freivalds' check cannot be re-used indefinitely. The most efficient approach would presumably be to train with very large batches than can then be verified simultaneously.
- Finally, the pre-computation techniques we employ for protecting input privacy do not apply for training, as the weights change after every processed batch. Moreover, Slalom does not try to hide the model weights from the untrusted processor, which might be a requirement for private training.

## 6 Conclusion

This paper has studied the efficiency of evaluating a DNN in a Trusted Execution Environment (TEE) to provide strong integrity and privacy guarantees. We explored new approaches for segmenting a DNN evaluation to securely outsource work from a trusted environment to a faster co-located but untrusted processor.

We designed Slalom, a framework for efficient DNN evaluation that outsources all linear layers from a TEE to a GPU. Slalom leverage Freivalds' algorithm for verifying correctness of linear operators, and additionally encrypts inputs with precomputed blinding factors to preserve privacy. Slalom can work with any TEE and we evaluated its performance using Intel SGX on various workloads. For canonical DNNs (VGG16, MobileNet and ResNet variants), we have shown that Slalom boosts inference throughput without compromising security.

Securely outsourcing matrix products from a TEE has applications in ML beyond DNNs (e.g., non negative matrix factorization, dimensionality reduction, etc.) We have also explored avenues and challenges towards applying similar techniques to DNN training, an interesting direction for future work. Finally, our general approach of outsourcing work from a TEE to a faster co-processor could be applied to other problems which have fast verification algorithms, e.g., those considered in (McConnell et al., 2011; Zhang et al., 2014).

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

## A  DETAILS ON INTEL SGX SECURITY

SGX enclaves isolate execution of a program from all other processes on a same host, including a potentially malicious OS. In particular, enclave memory is fully encrypted and authenticated. When a word is read from memory into a CPU register, a Memory Management Engine handles the decryption (Costan & Devadas, 2016).

While SGX covers many software and hardware attack vectors, there is a large and prominent class of side-channel attacks that it explicitly does not address (Costan & Devadas, 2016; Tramèr et al., 2017). In the past years, many attacks have been proposed, with the goal of undermining privacy of enclave computations (Xu et al., 2015; Brasser et al., 2017; Moghimi et al., 2017; Götzfried et al., 2017; Van Bulck et al., 2017; Lee et al., 2017). Most of these attacks rely on data dependent code behavior in an enclave (e.g., branching or memory access) that can be partially observed by other processes running on the same host. These side-channels are a minor concern for the DNN computations considered in this paper, as the standard computations in a DNN are data-oblivious (i.e., the same operations are applied regardless of the input data) (Ohrimenko et al., 2016).

The recent Spectre attacks on speculative execution (Kocher et al., 2018) also prove damaging to SGX (as well as to most other processors), as recently shown (Chen et al., 2018; Dall et al., 2018; Van Bulck et al., 2018). Mitigations for these side-channel attacks are being developed (Shinde et al., 2016; Shih et al., 2017; Chen et al., 2017; Intel Corp., 2018) but a truly secure solution might require some architectural changes, e.g., as in the proposed Sanctum processor (Costan et al., 2016).

We refrain from formally modeling SGX's (or other TEE's) security in this paper, as Slalom is mostly concerned with outsourcing protocols wherein the TEE acts as a client. We refer the interested reader to (Pass et al., 2017; Fisch et al., 2017; Subramanyan et al., 2017) for different attempts at such formalisms.

## B    FORMAL SECURITY DEFINITIONS AND PROOFS

We define a secure outsourcing scheme, between a client $\mathcal{C}$ and a server $\mathcal{S}$, for a DNN $F(x) : \mathcal{X} \to \mathcal{Y}$ from some family $\mathcal{F}$ (e.g., all DNNs of a given size). We first assume that the model $F$ is known to both $\mathcal{C}$ and $\mathcal{S}$:

**Definition B.1** (Secure Outsourcing Schemes). A secure outsourcing scheme consists of an offline preprocessing algorithm Preproc, as well as an interactive online protocol Outsource$\langle \mathcal{C}, \mathcal{S} \rangle$, defined as follows:

- $\texttt{st} \leftarrow \textsf{Preproc}(F, 1^\lambda)$: The preprocessing algorithm is run by $\mathcal{C}$ and generates some data-independent state $\texttt{st}$ (e.g., cryptographic keys or precomputed values to accelerate the online outsourcing protocol.)

- $\mathcal{Y} \cup \{\bot\} \leftarrow \textsf{Outsource}\langle \mathcal{C}(F, x, \texttt{st}), \mathcal{S}(F) \rangle$: The online outsourcing protocol is initiated by $\mathcal{C}$ with inputs $(F, x, \texttt{st})$. At the end of the protocol, $\mathcal{C}$ either outputs a value $y \in \mathcal{Y}$ or aborts (i.e., $\mathcal{C}$ outputs $\bot$).

The properties that we may require from a secure outsourcing scheme are:

- **Correctness:** For any $F \in \mathcal{F}$ and $x \in \mathcal{X}$, running $\texttt{st} \leftarrow \textsf{Preproc}(F, 1^\lambda)$ and $y \leftarrow \textsf{Outsource}\langle \mathcal{C}(F, x, \texttt{st}), \mathcal{S}(F) \rangle$ yields $y = F(x)$.

- **t-Integrity:** For any $F \in \mathcal{F}$, input $x \in \mathcal{X}$ and probabilistic polynomial-time adversary $\mathcal{S}^*$, the probability that $\tilde{y} = \textsf{Outsource}\langle \mathcal{C}(F, x, \texttt{st}), \mathcal{S}^*(F) \rangle$ and $\tilde{y} \notin \{F(x), \bot\}$ is less than $t$.

- **Input privacy:** For any $F \in \mathcal{F}$, inputs $x, x' \in \mathcal{X}$ and probabilistic poly-time adversary $\mathcal{S}^*$, the views of $\mathcal{S}^*$ in $\textsf{Outsource}\langle \mathcal{C}(F, x, \texttt{st}), \mathcal{S}^*(F) \rangle$ and $\textsf{Outsource}\langle \mathcal{C}(F, x', \texttt{st}), \mathcal{S}^*(F) \rangle$ are computationally indistinguishable.

- **Efficiency:** The online computation of $\mathcal{C}$ in Outsource should be less than the cost for $\mathcal{C}$ to evaluate $F \in \mathcal{F}$.

**Model Privacy.**    In some applications a secure outsourcing scheme may also require to hide the model $F$ from either $\mathcal{S}$ or $\mathcal{C}$ (in which case that party would obviously not take $F$ as input in the above scheme).

Privacy with respect to an adversarial server $\mathcal{S}^*$ (which Slalom does not provide), is defined as the indistinguishability of $\mathcal{S}^*$'s views in $\textsf{Outsource}\langle \mathcal{C}(F, x, \texttt{st}), \mathcal{S}^* \rangle$ and $\textsf{Outsource}\langle \mathcal{C}(F', x, \texttt{st}), \mathcal{S}^* \rangle$ for any $F, F' \in \mathcal{F}$.

As noted in Section 2.1, a meaningful model-privacy guarantee with respect to $\mathcal{C}$ requires that $\mathcal{S}$ first commit to a specific DNN $F$, and then convinces $\mathcal{C}$ that her outputs were produced with the same model as all other clients'. We refer the reader to Canetti et al. (2002) for formal definitions for such *commit-and-prove* schemes, and to Tramèr et al. (2017) who show how to trivially instantiate them using a TEE.

**Proof of Theorem 3.2.**    Let $\texttt{st} \leftarrow \textsf{Preproc}$ and $\textsf{Outsource}\langle \text{TEE}(F, x, \texttt{st}), \mathcal{S} \rangle$ be the outsourcing scheme defined in Figure 1 (right). We assume that all random values sampled by the TEE are produced by a secure cryptographically secure pseudorandom number generator (PRNG) (with elements in $\mathbb{S} \subseteq \mathbb{F}$ for the integrity-check vectors $s$ used in Freivalds' algorithm, and in $\mathbb{F}$ for the blinding vectors $r_i$).

We first consider integrity. Assume that the scheme is run with input $x_1$ and that the TEE outputs $y_n$. We will bound $\Pr[y_n \neq F(x_1) \mid y_n \neq \bot]$. By the security of the PRNG, we can replace the vectors $s$ used in Freivalds' algorithm by truly uniformly random values in $\mathbb{S} \subseteq \mathbb{F}$, via a simple hybrid argument. For the $i$-th linear layer, with operator $W_i$, input $x_i$ and purported output $y_i$, we then have that $y_i \neq x_i W_i$ with probability at most $1/|\mathbb{S}|^k$. By a simple union bound, we thus have that $\Pr[y_n \neq F(x_1)] \leq n/|\mathbb{S}|^k - \text{negl}(\lambda)$. Note that this bound holds even if the same (secret) random values $s$ are re-used across layers.

For privacy, consider the views of an adversary $\mathcal{S}^*$ when Slalom is run with inputs $x_1$ and $x_1'$. Again, by the security of the PRNG, we consider a hybrid protocol where we replace the pre-computed blinding vectors $r_i$ by truly uniformly random values in $\mathbb{F}$. In this hybrid protocol, $\tilde{x}_i = x_i + r_i$ is simply a "one-time-pad" encryption of $x_i$ over the field $\mathbb{F}$, so $\mathcal{S}^*$'s views in both executions of the hybrid protocol are equal (information theoretically). Thus, $\mathcal{S}^*$'s views in both executions of the original protocol are computationally indistinguishable. $\qquad\square$

**Proof of Corollary 3.3.** The outsourcing protocol between the remote client $\mathcal{C}$ and server $\mathcal{S}$ hosting the TEE is simply defined as follows (we assume the model belongs to $\mathcal{S}$):

- $\mathtt{st} \leftarrow \mathsf{Preproc}()$: $\mathcal{C}$ and the TEE setup a secure authenticated communication channel, using the TEE's remote attestation property. The TEE receives the model $F$ from $\mathcal{S}$ and initializes the Slalom protocol.

- $\mathsf{Outsource}\langle \mathcal{C}(x, \mathtt{st}), \mathcal{S}(F)\rangle$:
  - $\mathcal{C}$ sends $x$ to the TEE over the secure channel.
  - The TEE securely computes $y = F(x)$ using Slalom.
  - The TEE sends $y$ (and a publicly verifiable commitment to $F$) to $\mathcal{C}$ over the secure channel.

If the TEE is secure (i.e., it acts as a trusted third party hosted by $\mathcal{S}$), then the result follows. $\qquad\square$

## C  PERFORMANCE COMPARISON OF DNN OUTSOURCING SCHEMES

We provide a brief overview of the outsourcing approaches compared in Table 1. Our baseline runs a DNN in a TEE (a single-threaded Intel SGX enclave) and can provide all the security guarantees of an ML outsourcing scheme. On a high-end GPU (an Nvidia TITAN XP), we achieve over $50\times$ higher throughput but no security. For example, for MobileNet, the enclave evaluates 16 images/sec and the GPU 900 images/sec ($56\times$ higher).

SafetyNets (Ghodsi et al., 2017) and Gazelle (Juvekar et al., 2018) are two representative works that achieve respectively integrity and privacy using purely cryptographic approaches (without a TEE). SafetyNets does not hide the model from either party, while Gazelle leaks some architectural details to the client. The cryptographic techniques used by these systems incur large computation and communication overheads in practice. The largest model evaluated by SafetyNets is a 4-layer TIMIT model with quadratic activations which runs at about 13 images/sec (on a notebook CPU). In our baseline enclave, the same model runs at over 3,500 images/sec. The largest model evaluated by Gazelle is an 8-layer CIFAR10 model. In the enclave, we can evaluate 450 images/sec whereas Gazelle evaluates a single image in 3.5 sec with 300MB of communication between client and server.

**A Note on Energy Efficiency.** When comparing approaches with different hardware (e.g., our single-core CPU baseline versus Slalom which also uses a GPU), throughput alone is not the fairest metric. E.g., the baseline's throughput could also be increased by adding more SGX CPUs. A more accurate comparison considers the *energy efficiency* of a particular approach, a more direct measure of the recurrent costs to the server $\mathcal{S}$.

For example, when evaluating MobileNet or VGG16, our GPU draws 85W of power, whereas our baseline SGX CPU draws 30W. As noted above, the GPU also achieves more than $50\times$ higher throughput, and thus is at least $18\times$ more energy efficient (e.g., measured in Joules per image) than the enclave.

For Slalom, we must consider the cost of running both the enclave and GPU. In our evaluations, the outsourced computations on the GPU account for at most $10\%$ of the total running time of Slalom (i.e., the integrity checks and data encryption/decryption in the enclave are the main bottleneck). Thus, the power consumption attributed to Slalom is roughly $10\% \cdot 85\mathrm{W} + 90\% \cdot 30\mathrm{W} = 35.5\mathrm{W}$. Note that when not being in use by Slalom, the trusted CPU or untrusted GPU can be used by other tasks running on the server. As Slalom achieves $4\times$-$20\times$ higher throughput than our baseline for the tasks we evaluate, it is also about $3.4\times$-$17.1\times$ more energy efficient.

## D   Notation for Standard Linear Operators

Below we describe some common linear operators used in deep neural networks. For simplicity, we omit additive bias terms, and assume that convolutional operators preserve the spatial height and width of their inputs. Our techniques easily extend to convolutions with arbitrary strides, paddings, and window sizes.

For a fully-connected layer $f_{\text{FC}}$, the kernel $W$ has dimension $(h_{\text{in}} \times h_{\text{out}})$. For an input $x$ of dimension $h_{\text{in}}$, we have $f_{\text{FC}}(x) = x^\top W$. The cost of the layer is $h_{\text{in}} \cdot h_{\text{out}}$ multiplications.

A convolutional layer has kernel $W$ of size $(k \times k \times c_{\text{in}} \times c_{\text{out}})$. On input $x$ of size $(h \times w \times c_{\text{in}})$, $f_{\text{conv}}(x) = \texttt{Conv}(x; W)$ produces an output of size $(h \times w \times c_{\text{out}})$. A convolution can be seen as the combination of two linear operators: a "patch-extraction" process that transforms the input $x$ into an intermediate input $x'$ of dimension $(h \cdot w, k^2 \cdot c_{\text{in}})$ by extracting $k \times k$ patches, followed by a matrix multiplication with $W$. The cost of this layer is thus $k^2 \cdot h \cdot w \cdot c_{\text{in}} \cdot c_{\text{out}}$ multiplications.

A separable convolution has two kernels, $W_1$ of size $(k \times k \times c_{\text{in}})$ and $W_2$ of size $(c_{\text{in}} \times c_{\text{out}})$. On input $x$ of size $(h \times w \times c_{\text{in}})$, $f_{\text{sep-conv}}(x)$ produces an output of size $(h \times w \times c_{\text{out}})$, by applying a depthwise convolution $f_{\text{dp-conv}}(x)$ with kernel $W_1$ followed by a pointwise convolution $f_{\text{pt-conv}}(x)$ with kernel $W_2$. The depthwise convolution consists of $c_{\text{in}}$ independent convolutions with filters of size $k \times k \times 1 \times 1$, applied to a single input channel, which requires $k^2 \cdot h \cdot w \cdot c_{\text{in}}$ multiplications. A pointwise convolution is simply a matrix product with an input of size $(h \cdot w) \times c_{\text{in}}$, and thus requires $h \cdot w \cdot c_{\text{in}} \cdot c_{\text{out}}$ multiplications.

## E   Neural Network Details

Table 3 provides details about the two DNNs we use in our evaluation (all pre-trained models are taken from Keras Chollet et al. (2015)). We report top 1 and top 5 accuracy on ImageNet with and without the simple quantization scheme described in Section 3.1. Quantization results in at most a $0.5\%$ drop in top 1 and top 5 accuracy. More elaborate quantization schemes exist (e.g., Micikevicius et al. (2018)) that we have not experimented with in this work.

We report the number of model parameters, which is relevant to the memory constraints of TEEs such as Intel SGX. We also list the total size of the inputs and outputs of all the model's linear layers, which impact the amount of communication between trusted and untrusted co-processors in Slalom, as well as the amount of data stored in the TEE when using Freivalds' algorithm with preprocessing.

Table 3: **Details of models used in our evaluation.** Accuracies are computed on the ImageNet validation set. Pre-trained models are from Keras Chollet et al. (2015).

| Model | Accuracy | | Quantized | | Layers | Parameters (M) | Size of layer inputs/outputs (M) |
| | Top 1 | Top 5 | Top 1 | Top 5 | | | |
|---|---|---|---|---|---|---|---|
| VGG16 | 71.0 | 90.0 | 70.6 | 89.5 | 16 | 138.4 | 9.1 / 13.6 |
| VGG16 (no top) | - | - | - | - | 13 | 14.7 | 9.1 / 13.5 |
| MobileNet | 70.7 | 89.6 | 70.5 | 89.5 | 28 | 4.2 | 5.5 / 5.0 |
| MobileNet (fused) | - | - | - | - | 15 | 4.2 | 3.6 / 3.1 |
| ResNet 50 | 76.9 | 92.4 | 76.4 | 92.2 | 50 | 25.5 | 10.0 / 10.4 |

## F  Modular Arithmetic with Floating Point Operations

In this section, we briefly describe how Slalom performs modular arithmetic over a field $\mathbb{Z}_p$ in the TEE, while leveraging standard floating point operations to maximize computational efficiency. The main computations in the TEE are inner products over $\mathbb{Z}_p$ for Freivalds' check (a matrix product is itself a set of inner products).

Our quantization scheme (see Section 3.1) ensures that all DNN values can be represented in $\mathbb{Z}_p$, for $p \lesssim 2^{24}$, which fits in a standard float. To compute inner products, we first cast elements to doubles (as a single multiplication in $\mathbb{Z}_p$ would exceed the range of integers exactly representable as floats). Single or double precision floats are preferable to integer types on Intel architectures due to the availability of much more efficient SIMD instructions, at a minor reduction in the range of exactly representable integers.

In our evaluation, we target a soundness error of $2^{-40}$ for each layer. This leads to a tradeoff between the number of repetitions $k$ of Freivalds' check, and the size of the set $\mathbb{S}$ from which we draw random values. One check with $|\mathbb{S}| = 2^{40}$ is problematic, as multiplying elements in $\mathbb{Z}_p$ and $\mathbb{S}$ can exceed the range of integers exactly representable as doubles ($2^{53}$). With $k = 2$ repetitions, we can set $\mathbb{S} = [-2^{19}, 2^{19}]$. Multiplications are then bounded by $2^{24+19} = 2^{43}$, and we can accumulate $2^{10}$ terms in the inner-product before needing a modular reduction. In practice, we find that increasing $k$ further (and thus reducing $|\mathbb{S}|$) is not worthwhile, as the cost of performing more inner products trumps the savings from reducing the number of modulos.

## G  Results on a Standard CPU

For completeness, and to asses how our outsourcing scheme fairs in an environment devoid of Intel SGX's performance quirks, we rerun the evaluations in Section 4 on the same CPU but outside of SGX's enclave mode.

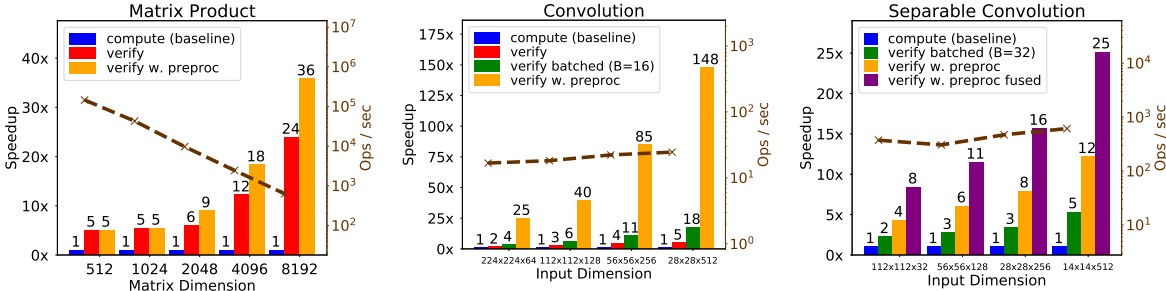

Figure 5: **Micro benchmarks on an untrusted CPU.** For three different linear operators, we plot the relative speedup of verifying a result compared to computing it. The dotted line in each plot shows the throughput obtained for computing the operation.

Figure 5 show the results of the micro-benchmarks for matrix multiplication, convolution and separable convolutions. In all cases, verifying a computation becomes 1-2 orders of magnitude faster than computing it as the outer dimension grows. Compared to the SGX benchmarks, we also see a much better viability of batched verification (we haven't optimized batched verifications much, as they are inherently slow on SGX. It is likely that these numbers could be improved significantly, to approach those of verification with preprocessing).

Figure 6 shows benchmarks for VGG16 and MobileNet on a single core with either direct computation or various secure outsourcing strategies. For integrity alone, we achieve savings up to $8.9\times$ and $19.5\times$ for MobileNet and VGG16 respectively. Even without storing any secrets in the enclave, we obtain good speedups using batched

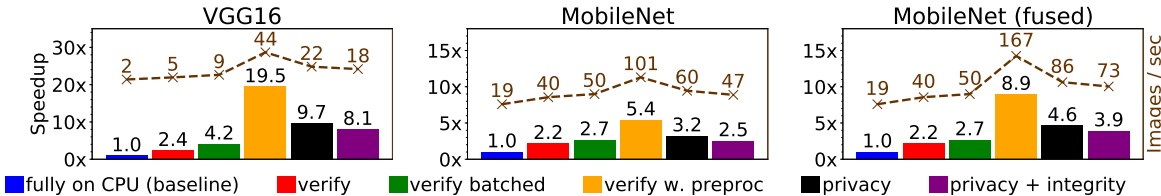

Figure 6: **Inference with integrity and/or privacy on an untrusted CPU.** We compare the baseline inference throughput (blue) to that obtained with "on-the-fly" integrity checks (red); batched integrity checks (green); integrity checks with precomputed secrets (yellow); privacy only (black); and privacy and integrity (purple). The fused MobileNet model has no intermediate activation for separable convolutions.

verification. As noted above, it is likely that the batched results could be further improved. With additional blinding to preserve privacy, we achieve speedups of $3.9\times$ and $8.1\times$ for MobileNet and VGG16 respectively.

# H PARALLELIZATION

Our experiments on SGX in Section 4 where performed using a single execution thread, as SGX enclaves do not have the ability to create threads. We have also experimented with techniques for achieving parallelism in SGX, both for standard computations and outsourced ones, but with little success.

To optimize for throughput, a simple approach is to run multiple forward passes simultaneously. On a standard CPU, this form of "outer-parallelism" achieves close to linear scaling as we increase the number of threads from $1$ to $4$ on our quad-core machine. With SGX however, we did not manage to achieve any parallel speedup for VGG16—whether for direct computation or verifying outsourced results—presumably because each independent thread requires extra memory that quickly exceeds the PRM limit. For the smaller MobileNet model, we get less than a $1.5\times$ speedup using up to $4$ threads, for direct computation or outsourced verification alike.

DNNs typically also make use of intra-operation parallelism, i.e., computing the output of a given layer using multiple threads. Our DNN library currently does not support intra-operation parallelism, but implementing a dedicated thread pool for SGX could be an interesting extension for future work. Instead, we evaluate the potential benefits of intra-op parallelism on a standard untrusted CPU, for our matrix-product and convolution benchmarks. We make use of Eigen's internal multi-threading support to speed up these operations, and custom OpenMP code to parallelize dot products, as Eigen does not do this on its own.

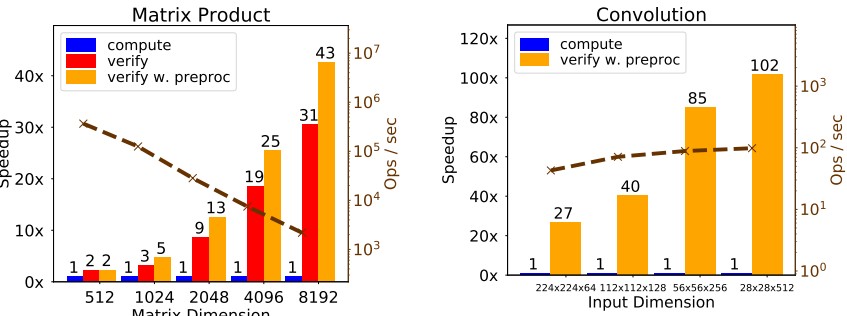

Figure 7: **Multi-threaded micro benchmarks on an untrusted CPU.** Reiterates benchmarks for matrix products and convolutions using $4$ threads.

Figure 7 shows the results using 4 threads. For convolutions, we have currently only implemented multi-threading for the verification with preprocessing (which requires only standard dot products). Surprisingly maybe, we find that multi-threading *increases* the gap between direct and verified computations of matrix products, probably because dot products are extremely easy to parallelize efficiently (compared to full convolutions). We also obtain close to linear speedups for verifiable separable convolutions, but omit the results as we currently do not have an implementation of multi-threaded direct computation for depthwise convolutions, which renders the comparison unfair. Due to the various memory-access overheads in SGX, it is unclear whether similar speedups could be obtained by using intra-op parallelism in an enclave, but this is an avenue worth exploring.

