# OpenReview forum: "Slalom: Fast, Verifiable and Private Execution of Neural Networks in Trusted Hardware"
_ICLR.cc/2019/Conference_

### Official Review · AnonReviewer3 · 2018-10-15
**strong paper, significant and solid results**

**Rating:** 9
**Confidence:** 4

**Review:**


Given the growing interest in building trust worthy and privacy protecting AI systems, this paper demonstrates a novel approach to achieve these important goals by allowing a trusted, but slow, computation engine to leverage a fast but untrusted computation engine. For the sake of protecting privacy, this is done by establishing an additive secret share such that evaluation on one part of the share is performed offline and the computation on the other part of the share is performed on the untrusted engine. To verify the correctness of the computation on the untrusted server, a randomized algorithm is used to sample the correctness of the results. Using these techniques, the authors demonstrate an order of magnitude speedup compared to running only on the trusted engine and 3-4 orders of magnitude speedup compared to software-based solutions.

Overall this is a strong paper which presents good ideas that have influence in ML and beyond. I appreciate the fact that the authors are planning to make their code publicly available which makes it more reproducible. Below are a few comments/questions/suggestions

1.	This papers, and other papers too, propose mechanisms to protect the privacy of the data while outsourcing the computation on a prediction task. However, an alternative approach would be to bring the computation to the data, which means performing the prediction on the client side. In what sense is it better to outsource the computation? Note that outsourcing the computation requires both complexity on the server side and additional computation on the client side (encryption & decryption).
2.	You present the limitations of the trust model of SGX only in the appendix while in the paper you compare to other techniques such as Gazzelle which have a different trust model and assumption. It makes sense to, at least, hint the reader on these differences.
3.	In section 2.2: “has to be processed with high throughput when available” is it high throughput that is required or low latency?
4.	In Section 4.3: in one of the VGG experiment you computed only the convolution layers which, as you say, are commonly used to generate features. In this case, however, doesn’t it make more sense that the feature generation will take place on the client side while only the upper layers (dense layers) will be outsourced?
5.	In section 4.3 “Private Inference” : do you include in the time reported also the offline preprocessing time? As far as I understand this should take the same amount of time as computing on the TEE.

---

> ### Author Response · Authors · 2018-11-24
> **Response to review**
>
> We thank the reviewer for the extremely positive review of our paper. As noted in our responses to the other reviewers, we have made some editorial changes to the paper (mainly moving some experiments from the Appendix into the main body), and we have included additional results for ResNet architectures as suggested by the second reviewer.
>
> To answer your insightful questions:
>
> 1. This is a great question, and such a system has recently been suggested in [1] (see below), which we now reference in our writeup. Probably the main differentiator is that in such a system, every single user requires trusted hardware (e.g., a recent Intel CPU), whereas in the cloud-outsourcing scheme which we and others have considered, only the server requires specialized hardware.
> Using Slalom's techniques in a client-side execution might work, but the downside is that our algorithm assumes that the untrusted host has knowledge of the computed model.
> One client-side application where Slalom might come in useful is for guaranteeing integrity in Federated Learning. Here, each client evaluates a model on their own data, and the server may need some guarantees that those computations were performed correctly.
>
> 2. We have added a note for this, thanks.
>
> 3. Our evaluation primarily focuses on throughput. Low latency is also desirable of course, but this might require using a different untrusted processor as GPUs tend to be outperformed by CPUs when operating on single-element batches. In our experiments, we evaluate batches of images (around 16 images) on the GPU, and then verify a single image at a time in the TEE. By replacing the GPU with a high-end CPU, we could thus also achieve low latency using Slalom.
>
> 4. VGG16 has the particularity that the feature extraction part (i.e., without the dense layers) makes up roughly 95% of the model's computation, but uses only maybe 5% of the weights (simply because VGG16's first dense layer is huge). So outsourcing only the dense layers would unfortunately leave the client to do most of the work.
> What we had in mind here was the use of VGG16's features as a building block for other models (e.g., object detection with SSD). We performed some preliminary experiments with SSD and a VGG16 backend and found that Slalom could also achieve around 10x speed improvements for such object-detection tasks.
>
> 5. Indeed, as you correctly note, the pre-computation takes the same time as the baseline computation in the TEE. The results in Section 4.3 do not include this pre-computations as we would of course not obtain any savings by doing so.
> A similar approach is taken by many Cryptographic approaches to secure outsourcing (of ML tasks or other computations). The rationale is that the pre-computation is data-independent and can be performed offline (e.g., during periods of low system use) and thus doesn't count towards the cost of the online throughput (or latency as discussed above).
>
> [1] MLCapsule: Guarded Offline Deployment of Machine Learning as a Service, Hanzlik et al., https://arxiv.org/abs/1808.00590

---

### Official Review · AnonReviewer1 · 2018-10-30
**Overall solid paper, would be stronger if it included a discussion of limitations of the approach**

**Rating:** 7
**Confidence:** 2

**Review:**

The authors propose a new method of securely evaluating neural networks. The approach builds upon existing Trusted Execution Environments (TEE), a combination of hardware and software that isolates sensitive computations from the untrusted software stack. The downside of TEE is that it is expensive and slow to run. This paper proposes outsourcing the linear evaluation portions of the DNN to an untrusted stack that's co-located with the TEE. To achieve privacy (i.e., the input isn't revealed to the untrusted evaluator), the approach adds a random number r to the input vector x, evaluates f(x+r) on the untrusted stack, then subtracts off f(r) from the output. This limits the approach to be applicable to only linear functions. To achieve integrity (verify the correctness of the output), the paper proposes testing with random input vectors (an application of Freivalds theorem, which bounds the error probability). The techniques for integrity and privacy works only on integer evaluations, hence the network weights and inputs need to be quantized. The paper tries to minimize degradation in accuracy by quantizing as finely as numerically allowable, achieving <0.5% drop in accuracy on two example DNNs. Overall, compared to full evaluation in a TEE, this approach is 10x faster on one DNN, and 40x to 64x faster on another network (depending on how the network is formulated).

Disclaimer: I am a complete outsider to the field of HW security and privacy. The paper is very readable, so I think I understand its overall gist. I found the approach to be novel and the results convincing, though I may be missing important context since I'm not familiar with the subject.

To me, the biggest missing piece is a discussion of the limitations of the approach. How big of a network can be evaluated this way? Is it sufficient for most common applications? What are the bottlenecks to scaling this approach?

It's also not clear why integrity checks are required. Is there a chance that the outsourcing could result in incorrect values? (It's not obvious why it would.)

Lastly, a question about quantization. You try to quantize as finely as possible (to minimize quantization errors) by multiplying by the largest power of 2 possible without causing overflow. Since quantization need to be applied to both input and network weights, does this mean that you must also bound the scale of the input? Or do you assume that the inputs are pre-processed to be within a known scale? Is this possible for intermediate outputs (i.e., after the input has been multiplied through a few layers of the DNN)?

Pros:
- Simple yet effective approach to achieve the goals laid out in the problem statement
- Clearly written
- Thorough experiments and benchmarks
- Strong results

Cons:
- No discussion of limitations
- Minor questions regarding quantization and size limits

Disclaimer: reviewer is generally knowledgeable but not familiar with the subject area.

---

> ### Author Response · Authors · 2018-11-24
> **Response to review**
>
> We thank the reviewer for the positive review and insightful comments. It is encouraging to hear that our paper was easy to read for someone outside the field of HW security and privacy.
>
> We have made some changes to our manuscript to better illustrate the limitations and scalability of our approach. We have moved our micro-benchmarks from the Appendix to the main body, as suggested by the second reviewer. These benchmarks show that as the computation performed in a layer gets larger (e.g., more channels in a convolution), the savings incurred by Slalom increase!
> In principle, Slalom scales linearly with the number of layers added to a model, so long as all the pre-computed values do not exceed SGX's memory limits. To illustrate, we have experimented with the family of ResNet architectures, that range from 18 to 152 layers (and 44MB to 230MB of weights). For five different models (18, 34, 50, 101 and 152 layers), Slalom provides large savings in throughput (4.4x-14.4x) and the savings tend to be larger for larger networks. We have added these results to our manuscript and we believe they illustrate Slalom's applicability and scalability to large models. In particular, the 152-layer ResNet model is among the deepest and most accurate models trained on ImageNet to date.
> Lastly, we have also moved a section from the Appendix to the main body wherein we discuss some limitations and challenges with extending Slalom to DNN training. The main issues here are with weights changing during training which hinders quantization, pre-computation of Freivalds' checks, as well as pre-computed blinding factors for privacy.
>
> Regarding the usefulness of integrity checks, these are also meant as a security guarantee. The threat model we consider here is that the server might *intentionally* compute incorrect values and send these back to the client. The SafetyNets paper by Ghodsi et al. contains a good discussion of reasons why a client may want integrity guarantees from the server. One example is a "model-downgrade attack", where the server runs a cheaper (i.e., smaller) model than advertised, to minimize costs. More generally, it is commonly agreed upon in the cryptographic community that privacy without integrity is an insufficient guarantee (e.g., by tampering with a client's results and observing the side effects, a server might later learn something about the client's data).
>
> For quantization, we simply assume that inputs are standard RGB images in the range [0, 255]. We then choose the quantization scales for inputs and weights so that none of the intermediate values in the network ever grow beyond p=2^23. The inputs of all layers after the first one are simply assumed to lie in the interval [-p/2, p/2].

---

### Official Review · AnonReviewer2 · 2018-11-02
**A nice systems approach to certain ML security problems with good performance**

**Rating:** 7
**Confidence:** 3

**Review:**

In this paper, the authors consider solving three ML security related challenges that would primarily arise
in the cloud based ML model. Namely, they consider the setting where a client wishes to obtain predictions
from an ML model hosted on a server, while being sure that the server is running the model they believe is being run
and without the server learning nothing about their input. Additionally, the server wishes for the user to learn
nothing about the model other than its output on the user's input. To solve this problem, the authors introduce a
new scheme for running ML algorithms in a trusted execution environment. The key idea is to oursource expensive
computation involved with forwarding images through a model to an untrusted GPU in a way that still allows for
the TEE to verify the integrity of the GPU's output. Because the authors' method is able to utilize GPU computing,
they achieve substantial speed-ups compared to methods that run the full neural network in trusted hardware.

Overall, I found the paper to be very well written and easy to digest, and the basic idea to be simple. The
authors strike a nice balance between details left to the appendix and the high level overview explained in
the paper. At the same time, the authors' proposed solution seems to achieve reasonably practicable performance
and provides a simple high-throughput solution to some interesting ML security problems that seems readily
applicable in the ML-as-a-cloud-service use case. I only have a few comments and feedback.

I would recommend the authors use the full 10 pages available by moving key results from the appendix to the main
text. At present, much of the experimental evaluation performed is done in the appendix (e.g., Figures 3 through
5).

The notation PR_{s \overset{s}{\gets}\mathbb{S}^{n}}[...] is not defined anywhere as far as I can tell
before its first usage in Lemma 2.1. Does this just denote the probability over a uniform random draw of
s from \mathbb{S}? If so, I might recommend just dropping the subscript: A, B, and C being deterministic
makes the sample space unambiguous. "negl(\lambda)" is also undefined.

In section three you claim that Slalom could be extended to other architectures like residual networks.
Can you give some intuition on how straightforward it would be to implement operations like concatenation
(required for DenseNets)? I would expect these operations could be implemented in the TEE rather than
on the coprocessor and then verified. However, the basic picture on the left of Figure 1 may then change,
as the output of each layer may need to be verified before concatenation? I think augmenting the right
of Figure 1 to account for these operations may be straightforward. It would be interesting to see
throughput results on these networks, particularly because they are known to substantially outperform
VGG in terms of classification performance.

---

> ### Author Response · Authors · 2018-11-24
> **Response to review**
>
> We thank the reviewer for the positive review and insightful comments.
>
> We followed the suggestion to make use of 10 pages of content (we originally found the ICLR Call for Papers to be somewhat unclear in this regard). We have moved parts of the Appendix into the main body, i.e., the SGX microbenchmarks as well as our discussion of challenges with extending Slalom to DNN training.
>
> We agree that the notation PR_{s \overset{s}{\gets}\mathbb{S}^{n}}[...] is overly verbose and we have removed the redundant in our updated manuscript. We have also added a definition for a negligible function.
>
> We followed the great suggestion to apply Slalom to more complex architectures. We have added experiments with ResNet models which make use of residual connections (handling concatenation layers would require similar changes to our framework). Extending Slalom's integrity checks (the left part in Figure 1) is quite trivial. The TEE simply applies Freivalds' algorithm to every linear operator and makes sure that it performs appropriate "book-keeping" of which layers' outputs correspond to which other layers' inputs. For privacy (the right part in Figure 1), things can get a bit more complicated as the TEE and GPU have to interact for each linear layer. For a residual layer, the TEE and GPU essentially run Slalom on both "paths" of the layer one after the other. The TEE saves intermediate results in its memory and then merges the results. The same would work for concatenation layers. Our results with ResNets are on par with those obtained with VGG16 and MobileNet. We tried different variants (18, 34, 50, 101 and 152 layers), and achieve 6.6-14.4x speedups for integrity and 4.4x-9.0x speedups with additional privacy.

---

### Author Response · Authors · 2018-11-24
**Summary of changes to the manuscript**

In response to the below reviews, we have made the following main changes to our paper:

- As suggested by the second reviewer, we have moved some of the content from the Appendix back to the main body. These include the microbenchmark results, as well as a discussion of the challenges in extending Slalom to DNN training.

- We have included new results for evaluating Slalom on ResNet models. This showcases Slalom's broad applicability to more complex models than the feed-forward architectures we had evaluated so far. We also illustrate Slalom's graceful scaling to very deep networks (e.g., the 152-layer ResNet with a 4.6x increase in private and verifiable throughput over the baseline).

- We have unified all figures to primarily display Slalom's *savings* over the baseline, rather than the raw throughput in images per second.

---

### Public Comment · ~Yu_Ding1 · 2018-12-21
**VIOLATES THE SGX THREAT MODEL**

As a security research scientist, I would like to say something from the security research perspective.

I've been working on SGX for a couple of years and keep maintaining rust-sgx-sdk and lot of SGX based stuffs for industrial use, including ML and DL.

I can see that the basic idea of this paper is to perform a transformation f(x) which holds f(x) + f(r) = f(x+r) and then do the transformation within SGX enclaves. It's confidentiality roots from r.

I would like to say it should not be a good idea -- it downgrades encryption from AES-256 to Cesear cipher. This is absolutely unacceptable in an SGX enclave. The basic principle of an SGX enclave is:

**Secret should be encrypted at least with AES-256-GCM before going out**

Obviously, **any** linear f(x) cannot satisfy this. And this is so much weak that cannot do any compliance stuffs. Nothing more to say about secret data such health data which has to comply with HIPAA.

To guarantee the security and perform the calculation on encrypted data, there are two ways:
(1) Do full-homomorphic encryption and then training/inferencing on the encrypted data.
(2) Use asymmetric encryption and conduct calculation on the encrypted data

I don't think this paper has any useful things to do with SGX because the proposed f could not be either (1) or (2). Every effort of such solutions would go to FHE and SMC, which is well-studied and has at least 10^4-10^5 performance overhead.

dingelish@gmail.com

---

> ### Author Response · Authors · 2018-12-21
> **Misunderstanding of our scheme**
>
> I don't quite follow your argument, which I think stems from a misunderstanding of our scheme.
>
> In our scheme, the linear function $f$ is computed outside the enclave. As you correctly mention, any data that leaves the enclave has to be properly encrypted and authenticated. So instead of outputting the plain input $x$, the enclave computes and outputs $z=x+r mod p$ where $r$ is a (pseudo)-random string generated in the enclave. If $r$ were purely random, this would be a one-time-pad, i.e., an information theoretically secure (but unauthenticated) encryption. In practice, $r$ is sampled from a cryptographic pseudorandom number generator based on AES, so $z$ is actually a stream cipher encryption of $x$ (which is provably secure assuming the PRG is secure, and definitely not a Caesar cipher...)
> Thus, as any $x$ that the enclave outputs is securely encrypted, the adversary provably learns no information about $x$.
>
> The remaining question is that of authenticated encryption (i.e., integrity). We sidestep the need for this by directly verifying integrity of the result of $f(z)$. So whether the adversary tampers with the encryption $z$, or with the computation of $f(z)$, the enclave will detect the tampering with overwhelming probability using Freivalds' algorithm.

---

> > ### Public Comment · ~Yu_Ding1 · 2018-12-22
> > **Disagree**
> >
> > The most valuable thing of SGX is its 0-cost AES256 ability.
> >
> > Let me use an example to show how the "random number" cannot protect your secret, no matter where it is.
> >
> > Assume you have a real random number $r$ and you send your message to Alice using the aforementioned algorithm with $r$. The ciphertext is considered to be crypto-unsafe because it does not defend against crypto analysis -- e.g. it does not change the statistical attribute of your plaintext, 'e' appears more frequently than other characters. An attacker can easily sniff the ciphertext and conduct a statistical attack and recover the $r$ which is not changed periodically -- your protocol does not support key rolling.
> >
> > That's the reason why your computer, your browser and your phone use TLS -- a transport layer security protocol which establishes connections using a complex algorithm and keeps keys rolling PERIODICALLY. The crypto algo would make the ciphertext changes significantly with only 1-bit change of plaintext and thus defend against crypto analysis attacks.
> >
> > In the SGX scenario, attackers are even more powerful than traditional communication scenario, where the attacker may intercept every byte. The attacker in an SGX scenario can 100% capture every bit of its input and output. A simple and fixed random number cannot defend from sophisticated crypto-analysis attacks. On the contrary, any encryption based on linear function is absolutely forbidden in laws -- because they are too weak.
> >
> > To sum up:
> > (1) no matter where the random number is, your proposed method is UNSAFE and IMPRACTICAL
> > (2) no matter where the encryption takes place, your proposed method is UNSAFE and IMPRACTICAL
> > (3) no matter what the linear function you use, your proposed method is UNSAFE and IMPRACTICAL
> >
> > YOU HAVE TO USE AT LEAST AES-256 ENCRYPTION AND PROPER KEY ROLLING MECHANISM TO ENCRYPT SENSITIVE DATA. THAT'S THE LAW!
> >
> > To be more clear, if you use this method on any secret data -- you violate the law!

---

> > > ### Author Response · Authors · 2018-12-22
> > > **The one time pad resists any cryptanalysis**
> > >
> > > This is incorrect.
> > > If we were sending $z=x+r$, then indeed this would be completely insecure.
> > >
> > > But $z=x + r mod p$ is provably secure against *any* cryptanalysis, if $r$ is uniformly random and used only once. This is the well known one-time pad encryption scheme: https://en.wikipedia.org/wiki/One-time_pad
> > > The one time pad is usually described over bits, i.e., $z = x XOR r = x + r mod 2$ where $x,r,z$ are bit-strings.
> > > But it is easy to show that when working over a larger field the encryption is still information theoretically secure.
> > >
> > > Moreover, our scheme never re-uses the same random $r$ more than once! That is, every time the enclave outputs an $x$, it uses a fresh random $r$. So the attacker cannot "recover" $r$ as it changes for every message. As I mentioned in my previous comment, we sample fresh $r$'s from a PRNG, so we're actually simply using a stream cipher (https://en.wikipedia.org/wiki/Stream_cipher).
> > >
> > > Stream ciphers are definitely not forbidden by law... For example, ChaCha20, one of the most efficient and popular stream ciphers today, is used in Google's implementation of ...TLS! https://en.wikipedia.org/wiki/Salsa20#ChaCha_variant

---

> > > > ### Public Comment · ~Yu_Ding1 · 2019-01-06
> > > > **Disagree**
> > > >
> > > > The non-colluding assumption is almost impossible in the real world, and this is why you and I are not on the same page. It is because an attacker could easily read the value of $u+r and $r when enclave passes them to the graphic card. So $r **HAS TO** be pre-calculated by one user remotely.
> > > >
> > > > So I agree that $r is a one-time pad and it masks secret data. But, $r here is a long-lasting one-time pad and its lifetime is much longer than crypto-safe one-time stuff because a client needs to precompute F(r) offline with poor computing performance. Obviously, if the client has the ability to compute F(r) efficiently and quickly, he can directly compute F(x). So in your proposed approach, one potential assumption is that the one who chooses $r does is a poor guy and can only compute slowly.
> > > >
> > > > So now it turns out to be another problem -- the lifetime of the one-time pad $r. The user needs to protect his $r in a rather long period of time in an untrusted environment -- contradicting with the target of SGX. This is the limitation and you mentioned it in the paper.
> > > >
> > > > The only solution may be -- to generate $r in user's enclave and calculate F(r) in an SGX enclave slowly -- using only 128MB memory on an i7 or E3 CPU. How many users would like to pay for this price? such as 1000 hrs of local CPU time in exchange for a short predict calculation?
> > > >
> > > > btw: the proposed solution relies on seedable PRNG. This is forbidden in secure SGX programming. You could find that there is no rand() function in Intel SGX SDK -- because SGX ONLY TAKES RANDOM NUMBERS FROM RDRAND. Using seedable rng in SGX is a really bad design.

---

> > > > > ### Author Response · Authors · 2019-01-06
> > > > > **Clarification on online/offline phases and remote clients**
> > > > >
> > > > > Our scheme requires no non-collusion assumption and no pre-computation by a remote client. Rather, the pre-computation is done by the server (in a trusted enclave).
> > > > >
> > > > > Let me clarify the setting we consider. We consider a remote client that interacts with an untrusted cloud server. This server hosts a trusted enclave that runs ML workloads on client data (we assume no enclave on the remote client's machine).
> > > > >
> > > > > When the client sends an input $x$ to the enclave, we can accelerate the private computation of $F(x)$ on the server by leveraging an untrusted GPU, as long as the enclave has previously computed $F(r)$.
> > > > > As you correctly observe, computing $F(r)$ takes as long as computing $F(x)$, but the advantage is that $F(r)$ is independent of any client data, and can thus be pre-computed by the server's enclave at any time (e.g., during periods of low client usage such as outside of business hours).
> > > > > Our protocol thus consists of an "offline phase", in which the server's enclave slowly pre-computes values $F(r)$ and securely stores them, followed by an "online phase" in which the enclave can very quickly and privately compute $F(x)$ on a remote client's data.
> > > > >
> > > > > Protocols with such dual offline and online phases are very common in the secure multiparty-computation literature (e.g., if you look at many of the MPC protocols proposed for the 2017 edition of the iDash competition you linked above, you'll find that many require an offline pre-computation phase).
> > > > > Requiring such an offline phase is a possible limitation of our protocol which we discuss and acknowledge in our paper. Whether the "price" of pre-computing on random values in an offline phase in order to accelerate the online phase is worth it will depend heavily on application requirements.
> > > > >
> > > > > We'd also like to note that the "lifetime" of the pre-computed values $F(r)$ need not be very long, as these values can be discarded as soon as they are used (e.g., the enclave could compute values $F(r)$ outside of business hours, and then consume them all a few hours later). Even if secret values were held on for a long time, we don't understand why this should contradict SGX's use-cases. This is exactly what SGX's sealing mechanism is for: https://software.intel.com/en-us/blogs/2016/05/04/introduction-to-intel-sgx-sealing
> > > > >
> > > > > Finally, the comment on seedable (cryptographic) PRNGs being forbidden or bad design in SGX is nonsense. The RDRAND instruction available on Intel processors simply calls a hardware-implemented PRNG that is seeded using a hardware entropy source. SGX enclaves can of course use RDRAND (or even better RDSEED) to generate a secure random seed to be used with any other software-implemented CSPRNG (the SGX developer guide even suggests this in its section on Random Number Generation). This is exactly what we do: we use RDRAND to seed an AES-based CSPRNG.

---

> > > > > > ### Public Comment · ~Yu_Ding1 · 2019-01-06
> > > > > > **Disagree**
> > > > > >
> > > > > > For the non-colluding assumption, I think we have nothing to discuss. It's your "assumption".
> > > > > >
> > > > > > Just tell me what you think about the following truth:
> > > > > >
> > > > > > There is no seedable pseudo-random number generation function in the whole Intel SGX SDK/PSW/Sample code. And in Intel SGX developer manual, it says
> > > > > > 'The tRTS provides a wrapper to the RDRAND instruction to generate a true
> > > > > > random number from hardware. The C/C++ standard library functions rand
> > > > > > and srand functions are not supported within an enclave because they only
> > > > > > provide pseudo random numbers. Instead, enclave developers should use the
> > > > > > sgx_read_rand function to get true random numbers.'
> > > > > >
> > > > > > Your implementation has a hidden assumption: PRNG in Intel SGX is allowed. However, it's not.

---

> > > > > > > ### Public Comment · ~Yu_Ding1 · 2019-01-06
> > > > > > > **HIGHLIGHT**
> > > > > > >
> > > > > > > Seedable PRNG vs RDRAND
> > > > > > >
> > > > > > > Seedable PRNG is not allowed in Intel SGX

---

> > > > > > > ### Author Response · Authors · 2019-01-06
> > > > > > > **PRNGs are allowed in SGX**
> > > > > > >
> > > > > > > I don't understand your comment about the non-colluding assumption. Non-collusion between which parties? As I mentioned in my previous comment, we make no such assumption.
> > > > > > >
> > > > > > > As for the PRNG, indeed, there is no seedable PRNG in the Intel Sample code (as far as I can tell). But there is also no sample code to implement ML models (for example) and that doesn't mean we aren't allowed to implement it ourselves.
> > > > > > >
> > > > > > > Of course, for most purposes, using Intel's sgx_read_rand will be simpler and faster than using your own PRNG. But for our particular application, using an explicit stateful PRNG makes things easier, without sacrificing any security.
> > > > > > > Finally, as I mentioned above, the Intel SGX developer guide explicitly says:
> > > > > > > "Software vendors that have an existing Pseudo-Random Number Generator (PRNG) should use the RDSEED instruction to benefit from the high-quality entropy source"
> > > > > > > (https://download.01.org/intel-sgx/linux-2.0/docs/Intel_SGX_Developer_Guide.pdf, page 26).
> > > > > > > So according to Intel, this is a perfectly natural and secure thing to do.
> > > > > > >
> > > > > > > *edit* It's of course worth mentioning that the rand and srand functions commonly available in C/C++ are not cryptographically secure PRNGs and are not available in SGX as they rely on randomness made available by the OS. We of course don't use these as they aren't secure. Instead, we use a cryptographically secure PRNG, such as AES run in CTR mode (https://en.wikipedia.org/wiki/Cryptographically_secure_pseudorandom_number_generator).

---

> > > ### Public Comment · ~Yu_Ding1 · 2018-12-22
> > > **Read materials**
> > >
> > > I strongly recommend you read the following links and see how to conduct analysis on secret data:
> > >
> > > (0) crypto attacks
> > > See what is crypto attacks and how they easily defeat attack linear encryption.
> > > https://en.wikipedia.org/wiki/Category:Cryptographic_attacks
> > >
> > > (1) iDash competition, focusing on the analysis of medical data. Each year the competition has 3 problems to solve. The research area is a combination of compliance and full-holomorphic encryption and performance.
> > > http://www.humangenomeprivacy.org/2018/competition-tasks.html
> > >
> > > (2) Platform Embedded Security Technology Revealed
> > > SGX is not what you have in mind -- it is designed and implemented for cloud and compliance, instead of client-side toy.
> > > https://link.springer.com/book/10.1007/978-1-4302-6572-6
> > >
> > > (3) AWS cloud computing compliance
> > > See what is law and how AWS comply with law
> > > https://aws.amazon.com/compliance/

---

### Meta-Review · Area_Chair1 · 2018-12-17
**A very interesting new contribution to privacy and neural networks**

**Confidence:** 5
**Recommendation:** Accept (Oral)

**Metareview:**

The authors propose a new method of securely evaluating neural networks.

The reviewers were unanimous in their vote to accept. The paper is very well written, the idea is relatively simple, and so it is likely that this would make a nice presentation.